
# The Influence of Turbulent Mixing on the Subsurface Chlorophyll Maximum

# Layer in the Northern South China Sea

Chenjing Shang[1], Changrong Liang[2, 3], Guiying Chen[2, 3], Yongli Gao[4]

[1]Shenzhen Key Laboratory of Marine Bioresources and Eco-environmental Science, College of Life Science and

Oceanography, Shenzhen University, Shenzhen 518060, PR China

[2]State Key Laboratory of Tropical Oceanography, South China Sea Institute of Oceanology, Chinese Academy of

Sciences, Guangzhou 510301, China

[3]Southern Marine Science and Engineering Guangdong Laboratory, Guangzhou 511458, China

[4]Equipment Public Service Center, South China Sea Institute of Oceanology, Chinese Academy of Sciences,

Guangzhou 510301, China

*Correspondence to*: Changrong Liang (crliang@scsio.ac.cn)

**Abstract.** We present observations from deployments of a turbulent microstructure instrument and a

CTD package in the northern South China Sea from April to May 2010. From these we determined

the turbulent mixing (dissipation rate $\varepsilon$ and diapycnal diffusivity $\kappa$), nutrients (phosphate, nitrate,

and nitrite), nutrient fluxes, and chlorophyll a (Chl-a) in two transects (A and B). Transect A was

located in region far away from the Luzon Strait where turbulent mixing in the upper 100 m was

weak ($\kappa \sim 10^{-6} - 10^{-4}$ m$^2$ s$^{-1}$). Transect B was located in region near the Luzon Strait where the

turbulent mixing in the upper 100 m was strong ($\kappa \sim 10^{-5} - 10^{-3}$ m$^2$ s$^{-1}$) due to the influence of the

internal waves originating from the Luzon strait and the water intrusion from the western Pacific. In

both transects, there was a thin subsurface chlorophyll maximum layer (SCML) (0.3-0.7 mg m$^{-3}$)

nested in the water column between ~50 and 100 m. The observations indicate that effects of

turbulent mixing on the distributions of nutrient and Chl-a were different in different transects. In

transect with weak turbulent mixing, nutrient fluxes induced by turbulent mixing transported

nutrients to the SCML but not to the upper water. Nutrients were sufficient to maintain a local SCML



phytoplankton population and the SCML remained compact. In transect with strong turbulent mixing,
nutrient fluxes induced by turbulent mixing transported nutrients not only to the SCML but also to
the upper water, which scatters the nutrients in the water column, and weakens and diffuses the
SCML.
**1. Introduction**
Subsurface chlorophyll maximum layers (SCMLs) are nearly ubiquitous in the ocean, which
have significant contribution to the water column biomass and primary production (Cullen, 2015).
The depth, thickness, and intensity are the three main factors to characterize the SCML, which are
mainly controlled by light attenuation and hydrological dynamic (Gong et al., 2014; G Li et al., 2012;
Taguchi, 1980). The hydrological dynamic affecting the SCML includes turbulent mixing, advection,
upwelling, mesoscale eddy, and circulation (Hu et al., 2014; Huisman et al., 2006; Kononen et al.,
1998; Ledwell et al., 2008; Lu et al., 2010; Vandevelde et al., 1987; Z K Wang and Goodman, 2010;
Williams et al., 2013a). Turbulent mixing is ubiquitous in the ocean and varies greatly in time and
space. In recent years, more and more studies focused on the influence of turbulent mixing on the
nutrient distribution and the nutrient supply of phytoplankton communities (Hales et al., 2009;
MacIntyre and Jellison, 2001; Schafstall et al., 2010; Sharples et al., 2007; Tanaka et al., 2012;
Tweddle et al., 2013; Williams et al., 2013b). For example, Hales et al. (2009) observed high vertical
turbulent nutrient fluxes in the euphotic zone at the New England shelf break front. The averaged
nitrate fluxes there were up to $6 \times 10^{-5}$ mmol N m$^{-2}$ s$^{-1}$, sufficient to support a net community
productivity of 30 mmol C m$^{-2}$ d$^{-1}$. Schafstall et al. (2010) reported the tidal-induced mixing and
diapycnal nutrient fluxes in the Mauritanian upwelling region. Nitrate fluxes at the base of the mixed
layer over the continental slope reached a mean value of $12 \times 10^{-2}$ µmol m$^{-2}$ s$^{-1}$. Study from



Tanaka et al. (2012) revealed that vertical turbulent fluxes sustain the high chlorophyll a (Chl-a)
region along the shelf break in the south eastern Bering Sea. Observation from Wang and Goodman
(2010) indicated that turbulent mixing modulates the thickness and intensity of SCML in Monterey
Bay. These studies indicated that nutrient flux induced by turbulent mixing is an important dynamic
factor for redistributing nutrients and supporting primary productivity.

The South China Sea (SCS) is a semi-closed basin characterized by abundant phytoplankton

and energetic internal waves. Phytoplankton is dominantly modulated by the nutrient distribution and
nutrient supply which are affected by turbulent mixing (Chen, 2005; Chen et al., 2004; Du et al.,
2017; Q P Li et al., 2016; Ning et al., 2004; Pan et al., 2012; Ryan et al., 2008; Wong et al., 2007;
Yin et al., 2001; Zhang et al., 2016), and the turbulent mixing is commonly related to internal waves
(Z Y Liu and Lozovatsky, 2012; Shang et al., 2017; St Laurent, 2008; Tian et al., 2009; Yang et al.,
2014). Numbers of internal waves propagate westward form the Luzon Strait into the northern SCS
and undergo nonlinear interactions during the propagation, providing a large amount of energy for
the turbulent mixing of the SCS (Xie et al., 2018; Zhao and Alford, 2006; Zhao et al., 2004). For
example, observation reported by St Laurent (2008) indicated that dissipation rates induced by
internal tidal waves in the shelf-break region can be $O(10^{-6})$ W kg$^{-1}$. Studies from Liu and
Lozovatsky (2012) showed that the level of dissipation at the north of 20°N is two times larger than
that to the south of 20°N. Measurements conducted by Shang et al. (2017) indicated that the spatial
distribution of turbulent mixing in the SCS is uneven. Strong turbulent mixing is mainly limited in
the region near the Luzon Strait. The unevenly distributed turbulent mixing might have different
effects on the distributions of nutrient and Chl-a in different regions of the SCS. However, few
studies have investigated the relationships of the turbulent mixing and distributions of nutrients and



Chl-a. Most studies involving nutrient supply and phytoplankton in the SCS focus on upwelling and
coastal currents (Gan et al., 2010; Han et al., 2013; Q P Li et al., 2016; K K Liu et al., 2002; J J
Wang and Tang, 2014). In this study, the microstructure, Chl-a, and nutrient data obtained from two
transects of the northern SCS are used to investigated the impact of turbulent mixing on the
distributions of nutrient and Chl-a.
**2. Data and method**

Physical and biogeochemical measurements were conducted from 26 April to 23 May 2010.

Figure 1 shows the stations from which the data we use in this study. These stations are divided into
two transects (A and B). Transect A was located in the region far away from the Luzon Strait and
transect B was located in the region near the Luzon Strait. Transect A includes six stations (A1-A6)
and transect B includes nine stations (B1-B9). Conductivity-temperature-depth (CTD) cast was made
at each station to collect hydrological and nutrient data. Temperature and salinity data were
documented with the Sea-Bird Electronic 911 plus. Water samples were collected with Niskin bottles
from different depths for nutrient extraction. The extraction method has been described by Hu et al.
(2014). Sea-water from each depth was pre-filtered through a Whatman GF/F and decanted into a
100 ml polycarbonate bottle, frozen immediately and stored at −20°C prior to analysis in the
laboratory. According to the standard colorimetric techniques (Kirkwood et al., 1996), the
concentrations of nitrate ($NO_3$), nitrite ($NO_2$), and phosphate ($PO_4$) were analyzed with a flow
injection analyzer (Quickchem 8500, Lachat Instruments, USA). Continuous time series of velocity
at 5 min intervals and 16 m vertical spacing between 38 and 982 m were obtained from a shipboard
acoustic Doppler current profiler (ADCP).

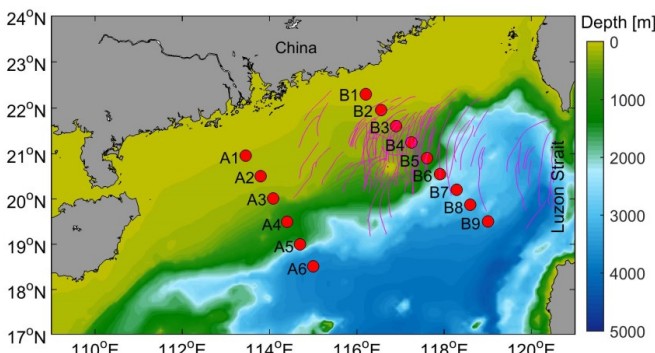


**Figure 1:** Bottom topography of the northern SCS with stations (circles) shown. Pink curves are

internal wave packets derived from satellite images by Zhao et al. (2004).

At all stations except station A4, turbulent microstructure data were collected with the
Turbulence Ocean Microstructure Acquisition Profiler (TurboMAP, Rockland Scientific Inc.) (Wolk
et al., 2002). TurboMAP is a quasi-free-falling instrument equipped with microstructure shear sensor,
temperature sensor, fluorescence sensor, pressure sensor, and turbidity sensor. The parameters
collected by TurboMAP include turbulent parameters (microscale velocity shear), bio-optical
parameters (fluorescence), and hydrographic parameters (conductivity, temperature, and depth). The
sinking rate of the profiler was 0.5-0.7 m s$^{-1}$. The Chl-a concentration from the fluorescence sensor
of TurboMap was calibrated by the bottle sampling. Dissipation rate ($\varepsilon$) was estimated with the
observed microscale velocity shear ($\partial u/\partial z$) using the following isotropic formula:

$$\varepsilon = 7.5\nu \left\langle \left(\frac{\partial u}{\partial z}\right)^2 \right\rangle = 7.5\nu \int_{k_1}^{k_2} \psi(k)\, dk, \qquad (1)$$

where $\nu$ is the kinematic viscosity, $<\,>$ denotes the spatial average, and $\psi(k)$ is the shear spectrum.
$k_1$ and $k_2$ are the integration limits. The lower integration limit $k_1$ is set to 1 cpm and the upper
limit $k_2$ is the highest wavenumber that is not contaminated by vibration noise.



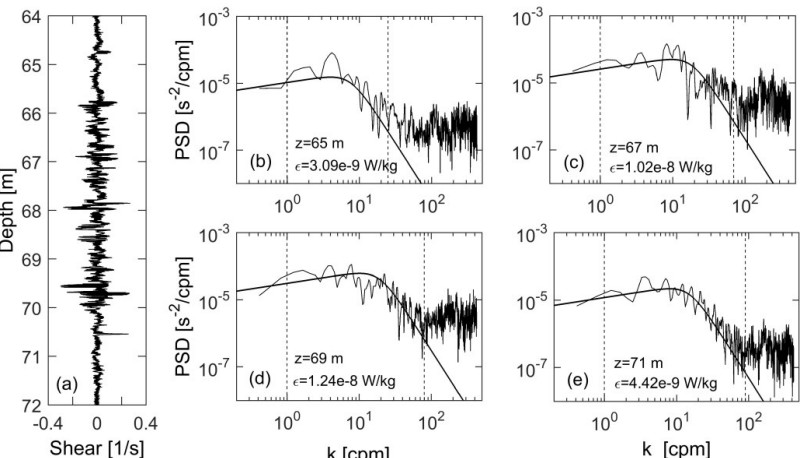


**Figure 2:** Examples of microscale velocity shear (a) at specified depth segments and the corresponding dissipation spectra (b-e). The smooth curves overlapping on the dissipation spectra are the Nasmyth spectra. The dashed vertical lines indicate the integration limit ranges.

Examples of microscale velocity shear and their corresponding dissipation spectra are shown in Figure 2. The dissipation spectra are approximately consistent with Nasmyth's spectra (Nasmyth, 1970) within the integration range (between the two dashed vertical lines). Distinct peaks associated with high wavenumbers (beyond the upper integration limit) were caused by instrument vibrations. The weak microscale velocity shears at depths of 65 m and 71 m correspond to weak dissipation ($\varepsilon \sim 10^{-9}$ W kg$^{-1}$) and strong microscale velocity shears at depths of 67 m and 69 m correspond to strong dissipation ($\varepsilon \sim 10^{-8}$ W kg$^{-1}$). Diapycnal diffusivity ($\kappa$) was calculated based on the dissipation rate and stratification (Osborn, 1980):

$$\kappa = \Gamma \frac{\varepsilon}{N^2}, \qquad (2)$$

where $\Gamma$=0.2 is the mixing efficiency (Gregg et al., 2018; Oakey, 1982) and $N^2$ is the squared buoyancy frequency. $N^2$ was calculated with the obtained temperature and salinity, which has a resolution of 1 m corresponding to the resolution of ε. The shear variance was calculated as





$S^2 = (\Delta \bar{U}/\Delta z)^2 + (\Delta \bar{V}/\Delta z)^2$ with $\Delta z$=16 m, where $\bar{U}$ and $\bar{V}$ are the respective zonal and meridional
components of the mean horizontal velocity obtained from the shipboard ADCP. The mean velocity
is averaged over the time intervals of the TurboMAP measurements.
**3. Results**
**3.1 Hydrographic condition**
Intrusion of water from the western Pacific can influence the water properties of the SCS.
Measurements and models (Shaw, 1991; Wu and Hsin, 2012) have confirmed that there is a strong
intrusion of water from the western Pacific into the SCS through the Luzon Strait. The *T-S* curves of
the two transects and the western Pacific are given in Figure 3. Data of the western Pacific (18.5$^o$
N-22.5$^o$ N, 124.5$^o$ E-128.5$^o$ E) were obtained from the World Ocean Database 2013. The *T-S* curve of
the western Pacific shows a reversed 'S' shape with one salinity minimum and one salinity
maximum. The maximum salinity layer (22.5-25.5 kg m$^{-3}$) corresponds to the high-salinity North
Pacific Tropical Water (NPTW) and the minimum salinity layer (25.5-27.5 kg m$^{-3}$) corresponds to
the low-salinity North Pacific Intermediate Water (NPIW) (Qu et al., 2000). NPTW mainly occupies
the water column in the upper 200 m and NPIW mainly occupies the water column below. The
salinity of transect B was close to the NPTW value in the maximum salinity layer. However, the
salinity of transect A was significantly smaller than that of NPTW. These observations indicate that
the influence of NPTW intrusion on the water properties of transect B was stronger than that of
transect A. Reversed trend was found in the minimum salinity layer. The minimum salinity in the
western Pacific is smaller than that in the two transects. Small salinity difference between transects A
and B and large salinity difference between the western Pacific and the two transects suggest that the
influence of NPIW intrusion on the water properties was weak in both transects.



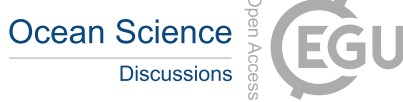

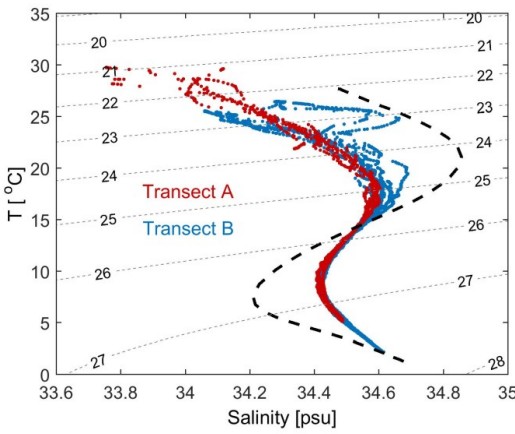


**Figure 3:** Relation of potential temperature versus salinity with potential density (unit in kg m$^{-3}$)

contours overlaid. The dashed curve shows the relation for potential temperature versus salinity of

the western Pacific for reference.

In addition to water intrusion, the SCS is also characterized by energetic internal waves. These

waves originate from the Luzon Strait and have strong impact on the velocity and temperature fields

of the SCS (Alford et al., 2015). Internal wave packets derived from satellite images by Zhao et al.

(2004) are shown in Figure 1 for reference. Transect A was located in the region where few internal

waves pass while transect B was located in the region where numbers of internal waves pass. To

further investigate the hydrologic condition of these two transects, we show the distributions of

temperature and salinity in Figure 4. The temperature in transect A shows a rapid temperature change

with increasing depth in the upper 100 m (Figure 4a) while the temperature in transect B remains

uniform in the upper 50 m and rapid temperature change mainly occur between 50 and 125 m

(Figure 4e). Similar distributions are observed in salinity (Figures 4b and 4f). Rapid salinity change

is found in the upper 50 m of transect A while the salinity in the upper 50 m of transect B remains

relatively uniform. In the upper 50 m, the temperature of transect A was higher than that of transect

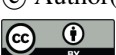

B but opposite in the salinity. Water intrusion and internal waves contribute to the difference in
hydrological conditions between the two transects. High-salinity NPTW intruded into the SCS
through the Luzon Strait and was mixed with the local water of transect B, which results in a high
salinity of transect B in the upper layer (Qu et al., 2000). Internal waves might play an important role
in mixing the local and invasive waters (Alford et al., 2015).

Using the temperature and salinity data, we estimate the stratification which is showed in

Figures 4c and 4g. A comparison of these two transects shows that the surface mixed layer was very
thin (<10 m) in transect A but thick (~45 m) in transect B. Below the surface mixed layer is a
thermocline with strong stratification. Here, we roughly define the top of the thermocline (that is, the
bottom of the surface mixed layer) as the depth at which $N^2=1 \times 10^{-4}$ s$^{-2}$ and the bottom of the
thermocline as the depth at which $N^2=2 \times 10^{-4}$ s$^{-2}$. The thermocline of transect A was mainly
limited in the upper 100 m while the thermocline of transect B was found at depth between 45 and
125 m. The thermocline stratification of transect A was stronger than that of transect B. Stratification
of transect A between 15 and 35 m reaches $7 \times 10^{-4}$ s$^{-2}$. The deep surface mixed layer and weak
thermocline stratification of transect B might be caused by the NPTW intrusion and internal waves.
Intrusion waters could change the salinity field by mixing with the local waters and internal waves
could enhance the turbulent mixing among the waters, which weakens the stratification of transect B.
Figures 4d and 4h show the distribution of squared shear for transects A and B, respectively. The
squared shear of transect B, evidently, was stronger than that of transect A, especially squared shear
at depth of 50-150 m where the level of $S^2$ was two to three times higher than that of transect A.
Strong shear of transect B results from the internal waves originating from Luzon Strait and has
important impact on the turbulent mixing.

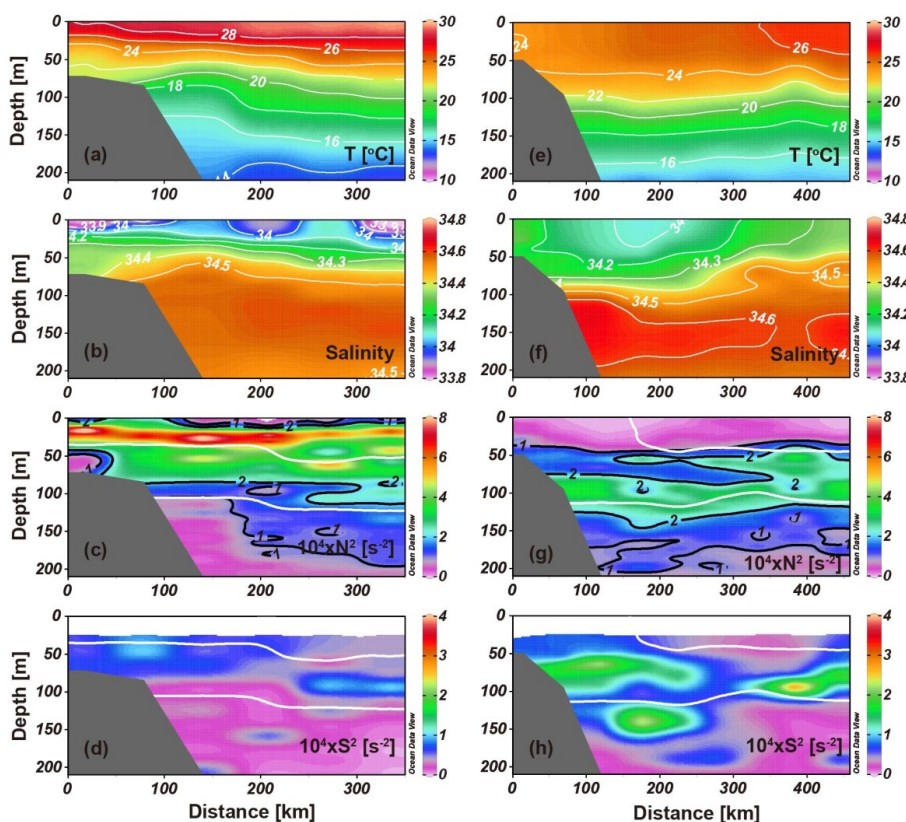

**Figure 4:** (Left) Distributions of (a) temperature, (b) salinity, (c) squared buoyancy frequency, (d) squared shear for transect A. (Right) The same as (left) but for transect B. Overlaid white lines in (c), (d), (g), and (h) are the boundaries of the subsurface chlorophyll maximum layer. The gray shading indicates the bathymetry.

**3.2 Distributions of $\varepsilon$ and $\kappa$**

Transect A has weak shear but strong stratification while transect B has strong shear but weak stratification. Stratification and shear are important conditions for turbulent mixing in the ocean (Liang et al., 2019b; MacKinnon and Gregg, 2003; 2005; Shang et al., 2017). To investigate the effects of stratification and shear on the turbulent mixing, we show the distributions of $\varepsilon$ and $\kappa$ in Figure 5. Data in the upper 10 m was removed due to contamination by the ship's wake and the





tilting of the TurboMAP profiler. In both transects, the upper 20 m was occupied by strong
dissipations with values of $\varepsilon$ reaching $O(10^{-8})$ W kg$^{-1}$ (Figures 5a and 5c). Dissipations of transect
B were stronger than that of transect A below 20 m. The averaged $\varepsilon$ below 20 m of transect B was
$1.92\times 10^{-8}$ W kg$^{-1}$, which is three times larger than that of transect B. Strong dissipations of
transect B were mainly caused by internal waves. Energetic internal waves propagate westward from
the Luzon Strait to the northern SCS and provide a large amount of energy for dissipation during
propagation (Alford et al., 2015; Z Y Liu and Lozovatsky, 2012; Shang et al., 2017). Diapycnal
diffusivity shows different distributions in transects A and B (Figures 5b and 5d). Diapycnal
diffusivity of transect A has a clear hierarchical structure. A weak diapycnal diffusivity layer with $\kappa$
of $10^{-7} - 10^{-6}$ m$^2$ s$^{-1}$ occupies the water column between ~20 and 50 m. This weak diapycnal
diffusivity layer was mainly due to the strong stratification between ~20 and 50 m (Figure 4c).
Strong stratification can suppress shear instability and weaken the diapycnal mixing (Liang et al.,
2019b; MacKinnon and Gregg, 2005; Polzin et al., 1996). Below the weak diapycnal diffusivity layer
is a slightly enhanced diapycnal diffusivity layer, occupying the water column between ~50 and 100
m. Values of $\kappa$ in this layer were $10^{-6} - 10^{-5}$ m$^2$ s$^{-1}$, almost one order of magnitude larger than
that of the upper layer. Diapycnal mixing below 100 was weak ($\kappa \sim 10^{-7} - 10^{-6}$ m$^2$ s$^{-1}$). There is no
hierarchical structure in the diapycnal mixing of transect B. Strong diapycnal mixing mainly
occurred in the upper 100 m and was one to three orders of magnitude larger than that of transect A.
The strong turbulent mixing in transect B was related to shear instability of internal waves which
depends on the shear and stratification. Generally, Richardson number $Ri = N^2/S^2$ is used to
assess the state of a water body and the water body is prone to shear instability in small Richardson
number (MacKinnon and Gregg, 2005). Transect A has strong stratification but weak shear (Figures




4c and 4d) while transect B has weak stratification but strong shear (Figures 4g and 4h), which
suggests that the water body in transect B is more prone to shear instability.A roughly estimation of
$Ri$ with 16 m shear and stratification indicates that the $Ri$ of transect B is smaller than the $Ri$ of
transect A (no shown).

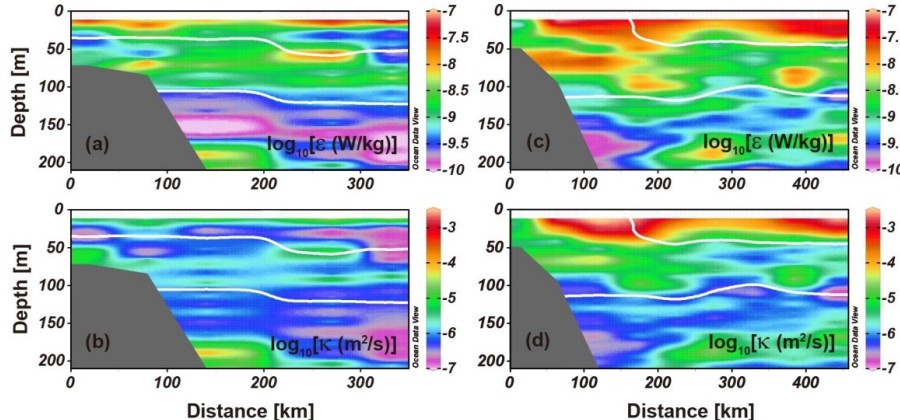


**Figure 5:** (Left) Distributions of (a) $\varepsilon$ and (b) $\kappa$ for transect A. (Right) The same as (left) but for

transect B. Overlaid white lines in each panel are the boundaries of the subsurface chlorophyll
maximum layer. The gray shading indicates the bathymetry.
**3.3 Distributions of Chl-a and nutrient concentrations**

Figures 6a and 6d show the distribution of Chl-a concentration for transects A and B,

respectively. The distribution of Chl-a concentration shows a sandwich structure in both transects. A
low Chl-a concentration layer with concentration lower than 0.25 mg m$^{-3}$ occupied the upper ~50 m.
A high Chl-a concentration layer with concentration higher than 0.25 mg m$^{-3}$ nested in the water
column between ~50 and 100 m. This layer is known as subsurface chlorophyll maximum layer.
Here we define the boundaries of SCML as the depth at which Chl-a concentration is equal to 0.25
mg m$^{-3}$. Below the SCML is another low Chl-a concentration layer with concentration lower than



0.25 mg m$^{-3}$. The SCML features of transect A are different from that of transect B. The SCML of
transect B occupies the whole water column on continental shelf (0 km<distance<170 km) while
transect A retains a hierarchical structure. The maximum Chl-a concentration was relatively stable
along transect A, while the maximum Chl-a concentration of transect B decreased at distance of
170-330 km. Overall, the SCML of transect A was more compact than that of transect B.

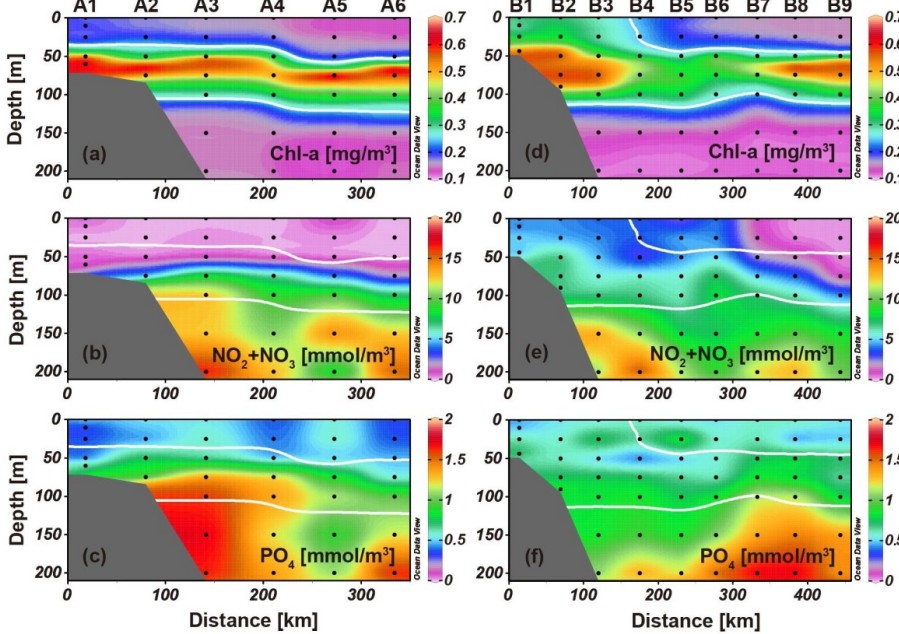


**Figure 6:** (Left) Distributions of (a) chlorophyll a (chl-a) concentration, (d) nitrate and nitrite
(NO$_2$+NO$_3$) concentration, (e) phosphate (PO$_4$) concentration for transect A. (Right) The same as
(left) but for transect B. Overlaid white lines in each panel are the boundaries of the subsurface
chlorophyll maximum layer. Solid dots indicate depths for nutrient collection. The gray shading
indicates the bathymetry.

Figures 6b and 6e show the distribution of nitrate and nitrite (NO$_2$+NO$_3$) concentration for
transects A and B, respectively. In transect A, NO$_2$+NO$_3$ concentration distributes evenly in





horizontal direction and has a clear nutricline. The water column in the upper 50 m was occupied by
$NO_2+NO_3$ concentration less than 2.5 mmol $m^{-3}$ and water column below 100 m was occupied by
$NO_2+NO_3$ concentration greater than 12.5 mmol $m^{-3}$. The water column between 50 and 100 m was
a nutricline in which the $NO_2+NO_3$ concentration increases rapidly with increasing depth, from
~2.5 mmol $m^{-3}$ at 50 m to ~12.5 mmol $m^{-3}$ at 100 m. The nutricline almost coincides with the
SCML. A different pattern is found in the distribution of transect B. The distribution of $NO_2+NO_3$
concentration was scattered and chaotic. No nutricline was found in this transect. The water column
in upper 75 m was occupied by $NO_2+NO_3$ concentration with values smaller than 7.5 mmol $m^{-3}$ and
water column below 75 m was occupied by $NO_2+NO_3$ concentration with values larger than 7.5
mmol $m^{-3}$. Overall, transect B had more $NO_2+NO_3$ than transect A above 75 m, but less $NO_2+NO_3$
than transect A below 75 m. Similarly, transect A also had a clear nutricline between 50 and 100 m
in the distribution of $PO_4$ concentration (Figure 6c) and no nutricline was found in the distribution
of $PO_4$ concentration of transect B (Figure 6f). Transect B had more $PO_4$ than transect A above 75
m, but less $PO_4$ than transect A below 75 m.
**4. Discussions**
Both transects have a high Chl-a concentration layer (SCML) nested in the water column
between ~50 and 100 m. However, the SCML of transect A was more compact than that of transect
B. In addition, the nutrient distributions ($NO_2+NO_3$ and $PO_4$) of transect B were more scattered and
chaotic than that of transect A. Turbulent mixing plays an important role in redistributing momentum,
heat, nutrients, and microorganisms in the ocean (Inall et al., 2001; Liang et al., 2019a; Shroyer et al.,
2010). To investigate the impact of turbulent mixing on the distribution of nutrients, we estimate the
nutrient flux induced by turbulent mixing, which is calculated as (Schafstall et al., 2010)



$$\Phi = -\kappa \frac{dC}{dz}, \qquad (3)$$

where $dC/dz$ is the vertical gradient of the dissolved nutrient concentration in the sample. To
calculate the nutrient flux, nutrient concentration was first interpolated onto the diapycnal diffusivity
grid. For simplicity, we designate the vertical gradient of $NO_2+NO_3$ concentration as $N_z$, the vertical
gradient of $PO_4$ concentration as $P_z$, the $NO_2+NO_3$ flux as $\Phi_N$, and the $PO_4$ flux as $\Phi_P$. The
distributions of nutrient flux are given in Figures 7a, 7b ,7e, and 7f. Nutrient fluxes in transect A
show a multi-layer structure (Figures 7a and 7b). A strong nutrient flux layer occupies the upper 20
m with $\Phi_N$ and $\Phi_P$ being $10^{-7}$ mmol m$^{-2}$ s$^{-1}$ and $10^{-8}$ mmol m$^{-2}$ s$^{-1}$, respectively. These strong
nutrient fluxes mainly resulted from strong turbulent mixing, as evidenced by the observations that
values of $\kappa$ were large (Figure 5b) but values of $N_z$ and $P_z$ were almost zero (Figures 7c and 7d) in
the upper 20 m. Lying below the strong nutrient flux layer is a weak nutrient flux layer ($\Phi_N \sim 10^{-8}$
mmol m$^{-2}$ s$^{-1}$ and $\Phi_P \sim 10^{-9}$ mmol m$^{-2}$ s$^{-1}$), which occupies the water column between ~20 and 50 m.
Weak nutrient fluxes in this layer were mainly due to the small vertical nutrient gradient (Figures 7c
and 7d) and weak turbulent mixing (Figure 5b). Most of $N_z$ were smaller than 0.1 mmol m$^{-4}$ (Figure
7c), $P_z$ smaller than 0.01 mmol m$^{-4}$ (Figure 7d), and $\kappa$ smaller than $5 \times 10^{-7}$ m$^2$ s$^{-1}$ (Figure 5b)
between ~20 and 50 m. Weak nutrient fluxes indicate that few nutrients were transferred upward
from deep layer. Below the weak nutrient flux layer, an enhanced nutrient flux layer ($\Phi_N \sim 10^{-7}$ mmol
m$^{-2}$ s$^{-1}$ and $\Phi_P \sim 10^{-8}$ mmol m$^{-2}$ s$^{-1}$) exists, occupying the water column between ~50 and 100 m. This
layer coincides with the nutricline (Figures 6b and 6c) and the SCML. Both the large vertical nutrient
gradient (Figures 7c and 7d) and strong turbulent mixing (Figure 5b) contribute to the strong nutrient
fluxes in this layer. Strong nutrient fluxes indicate that nutrients were transferred upward from the
deep layer through turbulent mixing. Nutrient fluxes below 100 m were weak due to the small
vertical nutrient gradient and weak turbulent mixing.

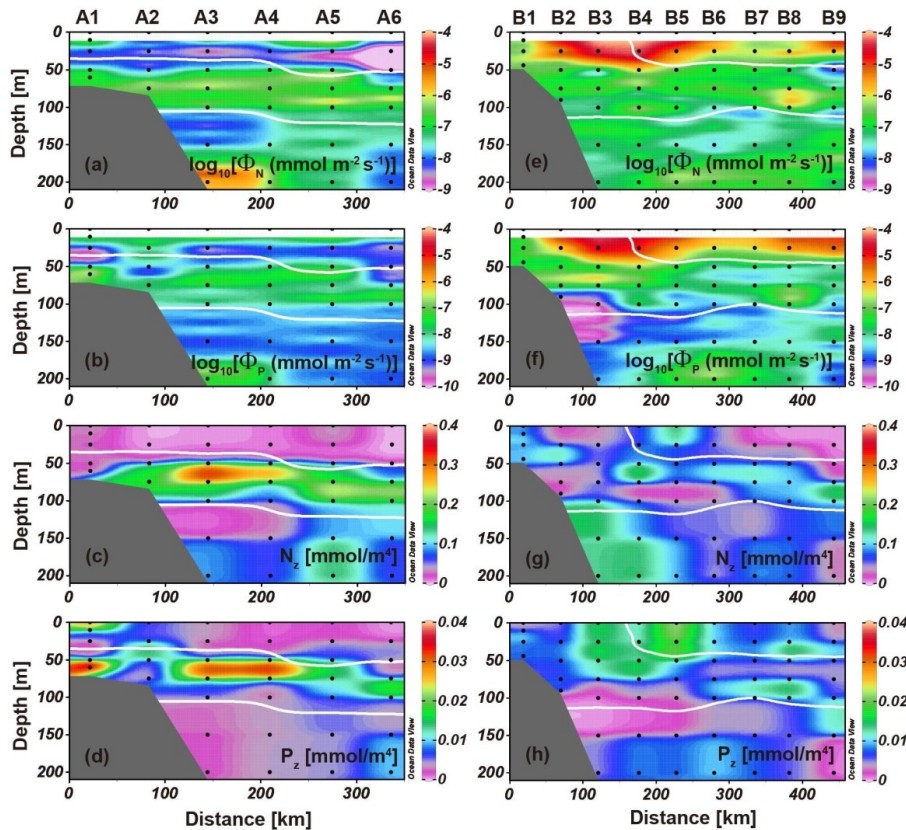


**Figure 7:** (Left) Distributions of (a) nitrate and nitrite flux ($\Phi_N$), (b) phosphate flux ($\Phi_P$), (c) vertical
gradient of nitrate and nitrite concentration ($N_z$), and (d) vertical gradient of phosphate concentration
($P_z$) for transect A. (Right) The same as (left) but for transect B. Overlaid white lines in each panel
are the boundaries of the subsurface chlorophyll maximum layer. Solid dots are depths for nutrient
collection. The gray shading indicates the bathymetry.

A different distribution was found in transect B. The upper 100 m was occupied by strong

nutrient fluxes and there was no multi-layer structure (Figures 7e and 7f). Values of $\Phi_N$ and $\Phi_P$
were one to three orders of magnitude larger than that of transect A. These strong nutrient fluxes
were mainly due to strong turbulent mixing, as evidenced by the observations that most values of $N_z$




and $P_z$ were smaller than 0.15 mmol m$^{-4}$ (Figures 7g and 7h) while values of $\kappa$ can be $O(10^{-4})$ m$^2$ s$^{-1}$
(Figure 5e). Strong nutrient fluxes indicate that nutrient transport in water column was strong and
nutrients were transported upward from the deep layer. It can be seen from Figures 6b, 6c ,6e, and 6f
that transect B has more nutrients than transect A in the upper 75 m but fewer nutrients than transect
A below 75 m. Strong nutrient fluxes induced by turbulent mixing also made the nutrient distribution
of transect B more scattered and chaotic than that of transect A. Next, we analyse the effect of
turbulent mixing on the Chl-a distribution.
In transect A, turbulent mixing in the upper 50 m was weak and few nutrients were transported
upward by the turbulent mixing. Nutrients was insufficient to maintain the phytoplankton population
in the upper 50 m, which exacerbates the deficiency of Chl-a. Note that surface phytoplankton bloom
earlier in the season might also cause the lack of nutrients in the surface waters. In the SCML,
nutrient fluxes induced by turbulent mixing continuously transport nutrients from deep layer to
SCML, which is sufficient to maintain a local SCML phytoplankton population and keep the SCML
compact (Figure 6a). Chl-a concentration below the SCML was low though nutrients were abundant.
This is mainly due to the lack of light in the deep layer. The effect of turbulent mixing on the Chl-a
distribution of transect B is different from that of transect A. On the continental shelf (0
km<distance<170 km), Chl-a and nutrients were distributed throughout the water column, and there
is no nutricline (Figures 6d and 6e). Strong turbulent mixing and its induced nutrient fluxes might be
important factors for the redistribution of nutrients and Chl-a on the continental shelf. Diapycnal
diffusivities in the upper 100 m were $10^{-5} - 10^{-3}$ m$^2$ s$^{-1}$ on the continental shelf, one to two orders
of magnitude larger than that of transect A. Strong nutrient fluxes induced by turbulent mixing
transport nutrients upward from the bottom, which contributes to maintain the phytoplankton





population throughout the water column. Away from the continental shelf (170 km<distance<300
km), SCML was also affected by the strong diapycnal mixing (Figure 6d). The SCML can be
distinguished, but it is not as compact as that of transect A. Chl-a within the SCML distribute more
evenly and the maximum Chl-a concentration was about two times lower than that of transect A.
Evenly distributed Chl-a and low maximum Chl-a concentration might be caused by the strong
turbulent mixing. Strong turbulent mixing transports nutrients from SCML to the upper water,
dispersing nutrients and reducing nutrient concentrations in the SCML. All these factors affect the
SCML phytoplankton population in transect B, which makes the SCML less compact and the
maximum Chl-a concentration lower than that of transect A. In deep-sea region (300
km<distance<450 km), few nutrients was found in the surface layer and the nutricline become
apparent (Figure 6e). This might be due to the weak turbulent mixing between ~50 and 100 m
(Figure 5d). At distance of 300-450 km, diapycnal diffusivity between ~50 and 100 m was
comparable to that of transect A and shows a hierarchical structure similar to transect A (Figure 5b).
The SCML remains compact and the maximum Chl-a concentration in SCML was comparable to
that of transect A.
In addition to turbulent mixing, upwelling is another factor affecting the distributions of
chlorophyll and nutrient (Q P Li et al., 2016). Unlike the turbulent mixing, the upwelling transports
nutrients upward through advection, $-WdC/dz$, where $W$ is the upwelling velocity. Spatial
distributions of curl-driven upwelling velocity and wind stress are shown in Figure 8. Upwelling
velocity and wind stress are from 3-day mean METOP-ASCAT data (https://coastwatch.pfeg.noaa.
gov/erddap/index.html). During the observation of transect A, the wind direction was generally south
on the west of the transect and was northeast on the east of the transect. There was strong curl-driven



upwelling in this transect and the upwelling at stations A3-A5 can be larger than $10^{-5}$ m s$^{-1}$. The
wind direction was generally east during the observation of transect B and the velocity field was
predominantly dominated by small downwelling. The effects of the strong upwelling on the
chlorophyll and nutrient distributions of transect A can be observed in Figures 6a-6c. Both the
SCML and nutricline were lifted up by the upwelling and the biggest uplift occurred at station
A3-A5 where the upwelling velocity was strongest. Evidence of uplift induced by upwelling was
also found in the distributions of temperature and salinity (Figures 4a and 4b). Both isotherm and
isohaline were lifted up by upwelling at distance between 100 and 300 km. These observations
suggest that the upwelling mainly affect the large scale distribution of nutrients and Chl-a rather than
the fine structure. Transect B was predominantly dominated by small downwelling and its effect on
the distributions of nutrients and Chl-a was weak. There is no good correlation between the
downwelling and the variations of the SCML and nutricline (Figures 6d-6f), which suggests that the
scattered distribution of nutrients and Chl-a in transect B is not due to the upwelling or downwelling.
The transportation induced by turbulent mixing plays an important role in the scattered distribution
of nutrients and Chl-a in transect B.

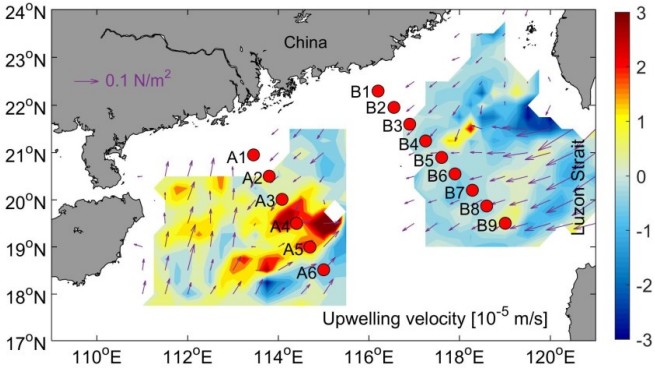


**Figure 8:** Spatial distributions of curl-driven upwelling velocity (color) and wind stress (vectors)



with stations (circles) shown. Upwelling velocity and wind stress are from 3-day mean
METOP-ASCAT data.
**5. Conclusions**
A field experiment has been conducted to study the effects of turbulent mixing on the
distributions of nutrients and Chl-a. Two transects were conducted during the experiment (transects
A and B). Transect A was located in the region far away from the Luzon Strait and transect B was
located in the region near the Luzon Strait where the turbulent mixing is strongly affected by internal
waves originating from the Luzon strait and water intrusion from the western Pacific. In both
transects, there is a high Chl-a concentration layer (SCML) nested in the water column between ~50
and 100 m. Turbulent mixing plays an important role in transporting nutrients from deep layer to the
SCML and maintaining the phytoplankton population. The effects of turbulent mixing on the
distributions of nutrients and Chl-a were different in different transects. In transect far away from the
Luzon Strait (transect A), the turbulent mixing was relatively weak and nutrients cannot be
transported to the surface layer by turbulent mixing. Nutrient fluxes induced by turbulent mixing is
sufficient to maintain a local SCML phytoplankton population but insufficient to replenish the
surface waters. The SCML remains compact in this transect. In transect near the Luzon Strait
(transect B), turbulent mixing was strong due to the influence of internal waves originating from the
Luzon strait and water intrusion from the western Pacific. Strong turbulent mixing transports
nutrients not only to the SCML but also to the upper waters above the SCML, which disperses
nutrient distribution and thus weakens and diffuses the SCML.
**Data availability**
The research data are available at Zenodo (http://doi.org/10.5281/zenodo.3864885).



**Author contributions**
Chenjing Shang, Guiying Chen, and Yongli Gao designed and carried out the experiments.
Changrong Liang prepared the manuscript with contributions from all co-authors.
**Competing interests**
The authors declare that they have no conflict of interest.
**Acknowledgments**
This work is supported by the National Key R&D Program of China under contract No.
2018YFA0902500; the National Natural Science Foundation of China under contract Nos 41706137,
41806033 and 41876023; the Natural Science Foundation of Guangdong Province of China under
contract No. 2017A030310332; State Key Laboratory of Tropical Oceanography, South China Sea
Institute of Oceanology, Chinese Academy of Sciences under contract No. LTO1909; the Natural
Science Foundation of SZU under contract Nos 2019078 and 860-000002110258; the Dedicated
Fund for Promoting High-Quality Economic Development in Guangdong Province (Marine
Economic Development Project) under contract No. GDOE[2019]A03; the Key Special Project for
Introduced Talents Team of Southern Marine Science and Engineering Guangdong Laboratory
(Guangzhou) under contract No. GML2019ZD0304; the Independent Research Project Program of
State Key Laboratory of Tropical Oceanography under contract No. LTOZZ1902. We thank all the
crew of the survey ship from the South China Sea Institute of Oceanology, Chinese Academy of
Sciences. We are very grateful to Professor Tan Yehui for her advice and nutrient data. The
numerical simulation is supported by the High Performance Computing Division and HPC managers
of Wei Zhou and Dandan Sui in the South China Sea Institute of Oceanology.

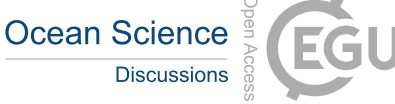

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
