# Peer review of "The Influence of Turbulent Mixing on the Subsurface Chlorophyll Maximum"

_Ocean Science, 2020_

## Short Comment (SC1) · 23 Jun 2020

As a small recommendation to improve this manuscript, I encourage the authors to make some changes to the figures to improve readability and avoid distortion of data:

1. Use of colourmaps. I urge the authors to avoid using the same jet style colourmap for several different variables, and instead use a range perceptually uniform colour-blind friendly colourmaps. Many options are available, including matplotlib's default colourmaps and those developed by Fabio Crameri for geosciences. An excellent set called cmocean has been developed specifically for oceanography by Kirsten Thyng et. al.

http://dx.doi.org/10.5670/oceanog.2016.66.

The cmocean colourmaps are described here with many good examples https://matplotlib.org/cmocean/. It appears that the authors are using Ocean Data Viewer to produce several of the figures, the cmocean colourmpas can easily be imported to this software https://github.com/kthyng/cmocean-odv

There has been considerable discussion, going back several years, on the misuses and dangers of the 'rainbow' or 'jet' colourmap. To avoid repetition, I refer readers to a blog post published on EGU 3 years ago on this topic

https://blogs.egu.eu/divisions/gd/2017/08/23/the-rainbow-colour-map/

In addition to making plots colourblind friendly and reducing visual distortion, using intuitive colourmaps for variables (blues for density, greens for chlorophyll) aids the reader in distinguishing subplots of different variables.

2. Reduce unnecessary repetition of plot objects, specifically colourbars and x and y axis labels. Where these are shared (e.g. figures 6) the figure can be made 'cleaner' by placing labels only on the outermost sets of plots and printing only one colourbar per variable. See contrast in figures attached to this comment.

3. Avoid printing variables names and units on plots. These can easily be included in the figure caption, or on the colourbar (see attached figures). This avoid cluttering the plot and obscuring data.

These comments are principally directed at figures 4-7. I have attached two figures made with synthetic data to represent temperature and chlorophyll at two different locations, as a simplified example parts of the authors' figures 4 and 6. Both figures use the same data. Contrast the use of variable-specific perceptually uniform colourmaps, sharing of axis tick labels and colourbars, and fewer graduations of the contour plot.

[Figure]

**Fig. 1.**

[Figure]

**Fig. 2.**

---

## Author Comment (AC1) · 27 Jun 2020

Thank you for your suggestion. We have modified and improved the Figures. Since the colourbar range of some Figures is different, we choose three colors in the colourbar to highlight their layered structure, for example, Figures 7a, 7b, 7e, and 7f.

[Figure]

**Fig. 1.** Figure 4: (Left) Distributions of (a) temperature, (b) salinity, (c) squared buoyancy frequency, (d) squared shear for transect A. (Right) The same as (left) but for transect B. Overlaid white and bla

[Figure]

**Fig. 2.** Figure 5: (Left) Distributions of (a) $\varepsilon$ and (b) $\kappa$ for transect A. (Right) The same as (left) but for transect B. Overlaid white lines in each panel are the boundaries of the subsurface chlorophyll max

**Fig. 3.** Figure 6: (Left) Distributions of (a) chlorophyll a (chl-a) concentration, (d) nitrate and nitrite (NO2+NO3) concentration, (e) phosphate (PO4) concentration for transect A. (Right) The same as (left)

[Figure]

**Fig. 4.** Figure 7: (Left) Distributions of (a) nitrate and nitrite flux (Φ_N), (b) phosphate flux (Φ_P), (c) vertical gradient of nitrate and nitrite concentration (N_z), and (d) vertical gradient of phosphate

---

## Referee Comment (RC1) · Anonymous Referee #1 · 6 Jul 2020

The article analyses two transects in the South China Sea conducted with CTD, turbulence profiles and ship board ADCP measurements. The authors focus on the transport mechanisms of nutrients through vertical turbulent mixing. The main results are strong differences between the two transects: the transects nearer to the Luzon Strait exhibits a much more patchy but also stronger turbulence and subsequently stronger vertical nutrient flux compared to the transect further away. The authors finish the article with this conclusion, which I find a bit weak and would expect more discussion, in the current form it is more a technical document describing a measurement. Possible questions which arise automatically could be: What are the consequences on biogeochemistry of this spatial inhomogeneity? Do satellite picture show also inhomogenities in chl-a? Is

the part nearer to the Luzon strait more/less productive? Maybe less/more fish catch? The introduction is lacking a section explaining the mechanisms of the evolution of chl-a. What are the sources/sinks, where are they and why are the authors at all interested in chl-a? Also the methodology needs some improvement, the authors do not describe the dates of the measurements, nor the meteorological situation. Also the processing is somewhat unclear, were the devices calibrated? What software was used to derive the dissipation rate? The computation of the fluxes does as well need a second look: As it is unclear what the time difference between the CTD and the turbulence profiles is, it is unclear how much error is induced by the time difference between the sampling. Transect B suggests by its patchiness a strong temporal and or spatial inhomogeneity, which has possibly a huge impact on the fluxes. The authors need to discuss this issue. On the other hand transect A has low turbulence O(1e-9 Wkg-1), a quick glance at the Buoyancy Reynolds number (eps/(N**2 nu )) at distance = 150 km, depth 50 m with eps~1e-9, N**2~4e-1 and nu~1e-6 gives a Reb = 2.5, suggesting values well below 10, in this region turbulent mixing is strongly damped or completely suppressed by stratification and fluxes are molecular. It is therefore necessary to compute Reb and to mark (or discard) regions of low Reb. Without being an natural English speaker, my impression is, that the English needs some improvement as well.

Despite these criticisms, this is a very valuable dataset and is worth publishing.

Detailed comments.

Figure 1: Add a subplot showing the region on an overview map, include in the subplot also the location of the western pacific mentioned in line 129.

Introduction: 30-51: Nutrient fluxes: where do the nutrients come from and were are they mixed/cponsumed/transported

81: Add date of last calibration

88-89: Add ADCP frequency and sampling intervals of the ADCP

100: Include a description of the software package used to calculate the dissipation rate.

Figure 2: Include the station number and date of measurement into the figure label

118: Detail in more detail how the interpolation of the high resolution T,S to eps was done

Data & Methods: It does not become clear when and how many profiles were taken. Was i.e. only one turbulence profile taken at the stations A1-A6 and B1-B9 or several? How were the meteorological conditions? Do tides play a role, was it spring/neap tide?

Figure 4: Add markers of the CTD/TurbMAP profiles

161-162: "Internal waves might play an important role in mixing the local and invasive waters (Alford et al., 2015).": This is not an result and better belongs into the introduction

171-172: What about local wind conditions? It could be argued that transect B was measured after a storm event, mixing the whole upper water column.

192: Figure 5 suggests O(10-7)

193-194: Transect B compared to transect B?

194-195: Is there evidence in measured data (i.e. ADCP) that internal waves are the main process, otherwise this is speculation (a reasonable though) and the sentence should be rephrased.

Figure 5b: One could argue that the profiles were taken with/without internal wave activity and thus creating the strong variability and patchiness of the data. Is there a way to estimate the internal wave activity during the profile? Add also markers for the locations of the profiles.

Figure 6: How do the oxygen profiles look like? Do they show similar patterns? 267:

Fluxes can be directed upwards/downwards, in the figure it is log10(flux). Add description of calculation

306: What does "maintaining" mean? The nutrient flux causes a growth of chl-a containing organisms, what processes cause a decay?

———————————————

---

## Referee Comment (RC2) · Anonymous Referee #2 · 7 Jul 2020

The Influence of Turbulent Mixing on the Subsurface Chlorophyll Maximum Layer in the Northern South China Sea by authors: Chenjing Shang et al. MS No.: os-2020-26

This manuscript considers biophysical implications of internal waves in the South China Sea using combined turbulence and nutrient data. While it is quite descriptive, the region and dynamics are very interesting and the results appear well-structured and can be considered in a wider context of other studies in the region. My general comment is it needs to clarify if it is a study about the generalities of SCM or is it primarily the biophysics of the South China Sea area? And the conclusion that a more turbulent region does more to diffuse and scatter a layer of increased productivity is not as clear

and strong as it could be.

In terms of language and grammar, while the text is readable and meaning is generally clear, there are many awkward or incorrect wordings/structures/spellings that a language edit would quickly clean up and make for a much easier read.

**Introduction**

This section consists of two quite dense paragraphs. I think these could be broken up and expanded on a. In the first paragraph I wanted to know more about horizontal variability. In the second I wanted to know more about the region – e.g. the actual location Figure is not referenced until the Methods section.

Line 52-on - as above it would be good to clarify if it is a study about the generalities of SCM or is it primarily the biophysics of the South China Sea area?

Line 62 – here and elsewhere values for dissipation rate are quoted. It would be good to get some sense of if these are average values or peak. This is especially true for internal wave driven processes which are typically sporadic – or at least spatiotemporally variable.

Line 71-73 It would benefit from a clear statement about the scientific question(s). Presently "In this study, the microstructure, Chl-a, and nutrient data obtained from two transects of the northern SCS are used to investigated the impact of turbulent mixing on the distributions of nutrient and Chl-a." seems quite vague. The material here has the building blocks of actual scientific questions, but they are not articulated. Is it all about internal waves? Are their horizontal nutrient gradients too? What are the temporal dynamics if it is internal wave mixing and this peaks for only a few hours every tidal cycle and shifting in/out of phase with daylight?

Also, some later material on mixing parameterisation (lines 210-) might be better here or methods.

**Methods**

Fig 1 – a zoom out would help locate for the unfamiliar. Also – how extensive are the results of Zhao et al. 2004 – do internal waves never penetrate to transect A?

Line 88 what frequency ADCP?

Line 94: Turbomap – I didn't think this was a Rockland Product.

Line 109-116 Turbulence analysis – there are some quite sophisticated and widely used methods for this. Were they used? Various references by Lueck, Wolk and colleagues look at vibration limit identification and replacement of the missing spectral region. Why was 1 m chosen as bin size?

Line 122 – "time interval of turbomap"? You mean the individual profiles of the full sampling period?

**Results**

There are many brief statements here that would better fit in the Discussion in a more expanded form. E.g. lines 161-162, 178-179

Line 190-191 – better placed into methods.

Lines 210-on The Gradient Ri gets introduced here which seems strange. It is not clear why it is required as there exists direct measures of turbulent mixing? Saying that, I do see the point about high vs low shear and stratification.

**Discussion**

This section is unstructured and would benefit from some clear themes building from the introduction. I think this needs significant work to give it structure and better bring together the results.

Lines 260-262 - Instead it starts with some Introductory material.

Line 260 "chaotic"? Is it actually chaotic or under-sampled? Under-sampling is inevitable in some situations so it is important to be clear. Do we have any sense of nutrient spatial variability beyond the transect data? e.g remote sensing or is the suggestion that these data are of limited use due to the subsurface biological processes?

Lines 359-375 - I don't really see the need for separate conclusions especially as they have much in common with abstract.

Can you plot biological production as a function of the nutrient flux in a way that shows how the two transects differ/compare?

Can you develop some kind of regime diagram where we have internal wave activity, nutrient availability and wind/upwelling and somehow present your findings for production in these terms?

Seasonality gets a minor mention but it would be useful to discuss more fully how these present results could/would translate through the annual cycle.

**Minor**

Potential temperature – some confusing notation and labelling – be good to be clear and use the accepted symbolic notation for potential temperature $\theta_0$.

---

## Editor Comment (EC1) · Ilker Fer (Editor) · 29 Jul 2020

Dear Chang-Rong Liang,

Thank you for your submission to the Ocean Science Discussion. While the observations you report can be of interest, I find the analysis short of qualifying a scientific article. The manuscript is highly descriptive, and no detailed analysis or convincing discussion is presented. The material would be more appropriate for a (very good) report or a data description paper. Following my comment and the comments from two reviewers, please post an authors' response that outlines how you intend to improve the study (e.g., which additional analysis and revisions you intend to do). At this stage

[Figure]

I do not encourage you to prepare a full revised version for submission to Ocean Science, but to summarize how you intend to proceed. Please allow me to decide based on your response.

The observations from two sections (one section with 6 stations and a second section with 9 stations) are presented as distance-depth distributions, like a data report: one figure for hydrography, one figure for the turbulence measurements and one figure for the water samples (Chl-a, nutrients and phosphate concentration). The next figure combines the diffusivity with concentration gradient to obtain the turbulent fluxes. The final figure implies curl-driven upwelling can be important, but it is introduced in passing and not a convincing analysis is presented. Overall, one section is turbulent and the other is not turbulent. Narrative repeatedly compares the two sections in various parameters.

Some minor comments: The observation period is given as approximately 1 month from late April to late May; however, which days the sections were collected are not stated. If the sections are separated by several weeks, the role of temporal variability could be discussed.

Descriptive occurrence of internal wave packets from satellite images (Zhao et al 2004) are from a different year. More convincing case could be made presenting actual observations from the cruise period.

Please make sure to use same units, for example nitrate fluxes are given in mmol N m-2 s-1 in line 43 and micromole m-2 s-1 in line 46.

Throughout, is it correct to refer to CTD as hydrological data?

What acoustic frequency is the ADCP?

Li 110, fine scale shear variance is introduced right after the microstructure shear variance. This can be confusing for the reader.

Please use standard notation for potential temperature and potential density anomaly.

[Figure]

Please do not use psu for salinity unit. Indicated that it is practical salinity, given on practical salinity scale (no units).

Li 129-130: salinity layers are identified in brackets in density ranges. This is confusing. At least clarify that the values are density.

Ri at 16-m vertical scale does not resolve the turbulence processes. It can be removed all together. Citing that previous literature reported Ri from 2m to 16 m vertical scale does not help (li 115).

In Fig 8, are the two separate parts around section A and B. I assume these are 3-day averages for days corresponding to the section time periods (i.e. you created a mosaic from two images). If so, please clarify in the caption and in the text.

---

## Author Comment (AC2) · 31 Jul 2020

The article analyses two transects in the South China Sea conducted with CTD, turbulence profiles and ship board ADCP measurements. The authors focus on the transport mechanisms of nutrients through vertical turbulent mixing. The main results are strong differences between the two transects: the transects nearer to the Luzon Strait exhibits a much more patchy but also stronger turbulence and subsequently stronger vertical nutrient flux compared to the transect further away. The authors finish the article with this conclusion, which I find a bit weak and would expect more discussion, in the current form it is more a technical document describing a measurement. Possible questions which arise automatically could be: What are the consequences on biogeochemistry of this spatial inhomogeneity? Do satellite picture show also inhomogeneities in chl-a? Is the part nearer to the Luzon strait more/less productive? Maybe less/more fish catch? The introduction is lacking a section explaining the mechanisms of the evolution of chl-a. What are the sources/sinks, where are they and why are the authors at all interested in chl-a? Also the methodology needs some improvements, the authors do not describe the dates of the measurements, nor the meteorological situation. Also the processing is somewhat unclear, were the devices calibrated? What software was used to derive the dissipation rate? The computation of the fluxes does as well need a second look: As it is unclear what the time difference between the CTD and the turbulence profiles is, it is unclear how much error is induced by the time difference between the sampling. Transect B suggests by its patchiness a strong temporal and or spatial inhomogeneity, which has possibly a huge impact on the fluxes. The authors need to discuss this issue. On the other hand transect A has low turbulence $O(1e-9 \text{ W kg}^{-1})$, a quick glance at the Buoyancy Reynolds number $(\text{eps}/(N^2 \text{nu}))$ at distance=150 km, depth 50 m with $\text{eps} \sim 1e^{-9}$, $N^2 \sim 4e^{-1}$ and $\text{nu} \sim 1e^{-6}$ gives a $\text{Reb} = 2.5$, suggesting values well below 10, in this region turbulent mixing is strongly damped or completely suppressed by stratification and fluxes are molecular. It is therefore necessary to compute Reb and to mark (or discard) regions of low Reb. Without being a natural English speaker, my impression is, that the English needs some improvement as well. Despite these

criticisms, this is a very valuable dataset and is worth publishing.

Responses: Thank you for your comments. Turbulent mixing can redistribute the nutrients and phytoplankton in the ocean, which would affect the distribution of microzooplankton and fish catch. The satellite picture also shows inhomogeneities in chl-a. Figure R1 shows the sea surface chlorophyll a during the survey obtained from ERDDAP (https://coastwatch.pfeg.noaa.gov/erddap/index.html). The concentration of sea surface chlorophyll a in the region of transect B was higher than that of transect A. To show the spatial distribution of sea surface chlorophyll a, we have replaced Figure 1 in the text with Figure R1 (line 89). Nutrient fluxes from below the euphotic zone are essential for phytoplankton primary production in the surface ocean. Unfortunately, we cannot quantify phytoplankton growth and microzooplankton grazing rates due to the lack of data. We have added content about the evolution of chl-a to the introduction (lines 35-41).

[Figure]

**Figure R1:** Spatial distribution of sea surface chlorophyll a with stations (circles) shown. The gray curves are the isobaths (unit in m). The dashed black box indicates the region where the temperature and salinity data of the western Pacific (19.5$^o$N-22$^o$N, 121.5$^o$E-123.5$^o$E) were obtained. Sea surface chlorophyll a is monthly MODIS-Aqua data (May 2010). The inset panel is a zoomed-out map with the South China Sea (SCS) and western Pacific (WP) shown. The black box in the inset panel indicates the observation area.

Transect A was conducted from 26 to 28 April 2010 and transect B was conducted from 22 to 23 May 2010. The weather was sunny and the wind was weak (<8 m s$^{-1}$) during the observation of the two transects (Figure R2). There was no storm during the observation. One CTD profile and one microstructure profile was conducted at each station. Microstructure profile was conducted right after the CTD profile. The observation time at each station is less than one hour over the continental shelf and less than two hours in deep sea. We believe that the turbulent field and the distribution of nutrients and chlorophyll have not changed greatly during this period. The devices had been calibrated before the cruise (January 2010). An integrated software application TMTools$^{TM}$ developed by Alec Electronics Co., Ltd. was used to derive the dissipation rate. We have added these details to the revised text (lines 98-109, 115-117, 123-124, and 133-135).

[Figure]

**Figure R2**: Wind amplitude for each station.

The formula of Buoyancy Reynolds number ($\varepsilon/(N^2 nu)$) is similar to that of diapycnal diffusivity ($\Gamma\varepsilon/N^2$). Diapycnal diffusivity is a variable to directly quantify the intensity of turbulent mixing. We have calculated the diapycnal diffusivity in the text. We believe that it is unnecessary to calculate the Buoyancy Reynolds number. Note that the colourmap of Figures 5-8 in the revised text was modified according to different variables, and a range more friendly colourmaps were used. The English has been improved by a natural English speaker (Figure R3).

**Invoice**

[Figure]

[Figure]

| Invoice | |
| --- | --- |
| Reference: | LE274096 |
| Order nr: | 254629 |
| Date: | Jul 20, 2020 |
| Currency: | USD |

**Product Information**

| Quantity | Description | Unit price | VAT rate | VAT amount | Total amount (usd) | Paid (usd) | Due |
| --- | --- | --- | --- | --- | --- | --- | --- |
| 1 | Language editing | 320.00 | 0.0% | 0.00 | 320.00 | 320.00 | 0.00 |
| | | | | | 320.00 | 320.00 | 0.00 |

**Customer Information**

| | |
| --- | --- |
| Name: | Guiying Chen |
| University: | South China Sea Institute of Oceanography |
| Department: | State Key Laboratory of Tropical Oceanography |
| Address: | Haizhu District, Xingang West Road, NO. 164 |
| Postal code / zip: | 510301 |
| City: | Guangzhou |
| State / province: | |
| Country: | China |

Total VAT amount: USD 0.00

VAT registration numbers: Austria (AT) U62029744, Belgium (BE) 454069965, Canada (CA) 899471825RT0001, Cyprus (CY) 99200008T, Czech Republic (CZ) 680459405, Denmark (DK) 17105779, Estonia (EE) 101100676, Finland (FI) 10123303, France (FR) 67390585230, Germany (DE) 172046177, Greece (GR) 999838602, Hungary (HU) 26952811, Ireland (IE) 9507113Q, Italy (IT) 00119309995, Luxembourg (LU) 21424724, Malta (MT) 18116507, Netherlands (NL) 801481247B01, Poland (PL) 5262786955, Portugal (PT) 980081963, Slovakia (SK) 4020110281, Slovenia (SI) 55336051, Spain (ES) A0063646D, Sweden (SE) 502048821801, Switzerland (CH) 494663, United Kingdom (GB) 494627212

**Figure R3:** Invoice of language editing.

Detailed comments.

Figure 1: Add a subplot showing the region on an overview map, include in the subplot also the location of the western pacific mentioned in line 129.

Responses: Thanks for your advice. We have expanded the map and selected another region (indicated by the dashed black box in Figure 1) for the temperature and salinity of the western Pacific (line 89).

Introduction: 30-51: Nutrient fluxes: where do the nutrients come from and are they mixed/consumed/transported

Responses: Nutrients in the ocean are usually distributed at deep layer and the seabed, while the nutrients in the upper ocean are relatively scarce due to consumption by phytoplankton. The role of nutrient flux is to transport nutrients from the deep and bottom layers to the upper layer, providing nutrients for the phytoplankton in the

upper ocean. We have made this clear in the revised text (lines 35-41).

81: Add date of last calibration

Responses: We have added the date to the revised text (line 109).

88-89: Add ADCP frequency and sampling intervals of the ADCP

Responses: The frequency of ADCP is 38 kHz. The sampling intervals were set to 5 min and 16 m bin size. We have clarified this in the revised text (lines 115-117).

100: Include a description of the software package used to calculate the dissipation rate.

Responses: Thanks for your advice. We have added a description of the software package in the revised text (lines 133-135).

Figure 2: Include the station number and date of measurement into the figure label

Responses: Thanks for your advice. We have added station number and date of measurement to the Figure label (line 156).

118: Detail in more detail how the interpolation of the high resolution T, S to eps was done

Responses: The CTD data were processed according to standard procedures as recommended by the manufacturer and bin averaged to 1-m resolution, corresponding to the resolution of the dissipation rate. We have added this to the revised text (lines 146-148).

Data & Methods: It does not become clear when and how many profiles were taken. Was i.e. only one turbulence profile taken at the stations A1-A6 and B1-B9 or several? How were the meteorological conditions? Do tides play a role, was it spring/neap tide?

Responses: One CTD profile and one microstructure profile were collected at each station. The weather was sunny and the wind was weak (<8 m s$^{-1}$) during the observation of the two transects. Figure R4 shows the barotropic tides obtained from the global inverse tide model (TPXO) at (20$^o$N, 116$^o$E) (Egbert and Erofeeva, 2002). Both transects are conducted in period of neap tide. The difference of turbulent mixing between transects A and B does not result from spring or neap tide. We have added these details in the revised text (lines 98-108 and 123-124).

Egbert, G. D. and Erofeeva, S. Y.: Efficient inverse modeling of barotropic ocean tides, J. Atmos. Ocean. Tech., 19, 183–204, 2002.

[Figure]

120 **Figure R4**: Time series of the barotropic tidal velocity predicted from TPXO 7.1. The two dashed red boxes indicate the periods of transects A and B.

Figure 4: Add markers of the CTD/TurbMAP profiles

Responses: Thanks for your advice. We have added station labels to the Figure (line 217).

125 161-162: "Internal waves might play an important role in mixing the local and invasive waters (Alford et al., 2015).": This is not an result and better belongs into the introduction.

Responses: Thanks for your advice. We have moved it to the introduction (line 62).

171-172: What about local wind conditions? It could be argued that transect B was

130 measured after a storm event, mixing the whole upper water column.

Responses: There was no storm during the observation. The weather was sunny and the wind was weak ($<8$ m s$^{-1}$) during the observation of the two transects. We have described the local wind conditions in the revised text (lines 103-106).

192: Figure 5 suggests $O(10^{-7})$

135 Responses: Most of values were less than $10^{-7}$ W/kg, so $O(10^{-8})$ W/kg is reasonable.

193-194: Transect B compared to transect B?

Responses: Thank you for your reminding. We have corrected the error in the revised text (line 229).

194-195: Is there evidence in measured data (i.e. ADCP) that internal waves are the

140 main process, otherwise this is speculation (a reasonable though) and the sentence should be rephrased.

Responses: Thanks for your comment and advice. No evidence can be found from the ADCP data due to the short-term observation. We have rephrased the sentence in the revised text (lines 229-231).

145 Figure 5b: One could argue that the profiles were taken with/without internal wave activity and thus creating the strong variability and patchiness of the data. Is there a way to estimate the internal wave activity during the profile? Add also markers for the locations of the profiles.

Responses: Thanks for your comment and advice. We cannot estimate the internal

150 wave activity due to the limit of velocity data. However, we can roughly determine whether the internal wave is active from the shear and stratification. Generally, the shear is relatively strong in region where the internal wave is active, and strong shear is likely to cause shear instability and weaken the stratification. As one can see from Figure 5 in the revised text that shear in transect B was evidently stronger than that of

155 transect A, and stratification in transect B was weaker than that of transect A. We have added station labels to the Figure 6 in the revised text.

Figure 6: How do the oxygen profiles look like? Do they show similar patterns? 267: Fluxes can be directed upwards/downwards, in the figure it is log10 (flux). Add description of calculation

160 Responses: Sorry, we have no oxygen data. The flux is calculated from Eq. 3 and then the absolute value is taken. We have explained this in the revised text (lines 299-300).

306: What does "maintaining" mean? The nutrient flux causes a growth of chl-a containing organisms, what processes cause a decay?

Responses: "maintaining" means "Provide nutrients for phytoplankton". We have

165 replaced "maintaining" with "support" in the revised text. Chl-a generally decays in the absence of nutrients or at high microzooplankton grazing rates.

170

---

## Author Comment (AC3) · 31 Jul 2020

The Influence of Turbulent Mixing on the Subsurface Chlorophyll Maximum Layer in the Northern South China Sea by authors: Chenjing Shang et al. MS No.: os-2020-26

This manuscript considers biophysical implications of internal waves in the South China Sea using combined turbulence and nutrient data. While it is quite descriptive, the region and dynamics are very interesting and the results appear well-structured and can be considered in a wider context of other studies in the region. My general comment is it needs to clarify if it is a study about the generalities of SCM or is it primarily the biophysics of the South China Sea area? And the conclusion that a more turbulent region does more to diffuse and scatter a layer of increased productivity is not as clear and strong as it could be.

In terms of language and grammar, while the text is readable and meaning is generally clear, there are many awkward or incorrect wordings/structures/spellings that a language edit would quickly clean up and make for a much easier read.

Responses: Thanks for your comment and advice. The study is primarily the biophysics of the South China Sea area. We have revised the article carefully, especially the discussion section, as suggested by the reviewers. In addition, the colourmap of Figures 5-8 in the revised text was modified according to different variables, and a range more friendly colourmaps were used. The English has been improved by a natural English speaker (Figure R1).

**Invoice**

[Figure]

[Figure]

| Invoice | | | | | | | |
|---|---|---|---|---|---|---|---|
| Reference: | LE274096 | | | | | | |
| Order nr: | 254629 | | | | | | |
| Date: | Jul 20, 2020 | | | | | | |
| Currency: | USD | | | | | | |

**Product Information**

| Quantity | Description | Unit price | VAT rate | VAT amount | Total amount (usd) | Paid (usd) | Due |
|---|---|---|---|---|---|---|---|
| 1 | Language editing | 320.00 | 0.0% | 0.00 | 320.00 | 320.00 | 0.00 |
| | | | | | 320.00 | 320.00 | 0.00 |

**Customer Information**

| | |
|---|---|
| Name: | Guiying Chen |
| University: | South China Sea Institute of Oceanography |
| Department: | State Key Laboratory of Tropical Oceanography |
| Address: | Haizhu District, Xingang West Road, NO. 164 |
| Postal code / zip: | 510301 |
| City: | Guangzhou |
| State / province: | |
| Country: | China |

Total VAT amount: USD 0.00

VAT registration numbers: Austria (AT) U62029744, Belgium (BE) 454069965, Canada (CA) 899471825RT0001, Cyprus (CY) 99200008T, Czech Republic (CZ) 680459405, Denmark (DK) 17105779, Estonia (EE) 101100676, Finland (FI) 10123303, France (FR) 67390585230, Germany (DE) 172046177, Greece (GR) 999838602, Hungary (HU) 26952811, Ireland (IE) 9507113Q, Italy (IT) 00119309995, Luxembourg (LU) 21424724, Malta (MT) 18116507, Netherlands (NL) 801481247B01, Poland (PL) 5262786955, Portugal (PT) 980081963, Slovakia (SK) 4020110281, Slovenia (SI) 55336051, Spain (ES) A0063646D, Sweden (SE) 502048821801, Switzerland (CH) 494663, United Kingdom (GB) 494627212

**Figure R1:** Invoice of language editing.

**Introduction**

This section consists of two quite dense paragraphs. I think these could be broken up and expanded on a. In the first paragraph I wanted to know more about horizontal variability. In the second I wanted to know more about the region – e.g. the actual location Figure is not referenced until the Methods section.

Responses: Thanks for your comment and advice. We have broken the introduction into four paragraphs. More details about the observational region are added and the location Figure is referenced in the introduction (lines 29-86).

Line 52-on - as above it would be good to clarify if it is a study about the generalities of SCM or is it primarily the biophysics of the South China Sea area?

Responses: Thanks for your comment. The study is primarily the biophysics of the northern South China Sea. We've clarified this in the revised text (lines 82-83).

35 Line 62 – here and elsewhere values for dissipation rate are quoted. It would be good to get some sense of if these are average values or peak. This is especially true for internal wave driven processes which are typically sporadic – or at least spatiotemporally variable.

Responses: Thanks for your advice. More details have been added to the revised text

40 (lines 58-69).

Line 71-73 It would benefit from a clear statement about the scientific question(s). Presently "In this study, the microstructure, Chl-a, and nutrient data obtained from two transects of the northern SCS are used to investigated the impact of turbulent mixing on the distributions of nutrient and Chl-a." seems quite vague. The material

45 here has the building blocks of actual scientific questions, but they are not articulated. Is it all about internal waves? Are their horizontal nutrient gradients too? What are the temporal dynamics if it is internal wave mixing and this peaks for only a few hours every tidal cycle and shifting in/out of phase with daylight? Also, some later material on mixing parameterisation (lines 210-) might be better here or methods.

50 Responses: Thanks for your comment and advice. In the study, we investigate the effect of vertical turbulent mixing on the vertical distribution of nutrients and chlorophyll. Turbulent mixing here is the result of the effects of internal waves with various frequencies and wavelengths. Without time series data, we cannot explore how the internal waves affect the vertical turbulent mixing at different periods and

55 locations. We are also unable to explore the effects of horizontal transport on the distribution of nutrients and chlorophyll due to the lack of horizontal turbulent mixing and horizontal flow data. We have added more details about our research in the introduction (lines 82-86). We have removed the mixing parameterization since it is not really relevant to the content of the article.

60 **Methods**

Fig 1 – a zoom out would help locate for the unfamiliar. Also – how extensive are the results of Zhao et al. 2004 – do internal waves never penetrate to transect A? Line 88 what frequency ADCP?

Responses: Responses: Thanks for your advice. We have embedded a zoomed-out
map in Fgiure 1. The internal wave packets in Zhao et al. 2004 are extracted from satellite images acquired from 1995 to 2001. Their results are extensive in the northern SCS. Internal wave packets propagate westwards to the continental shelf and dissipate there. Almost no internal wave packets penetrate to transect A. The ADCP we used is a 38 kHz ADCP. We have added the frequency to the revised text (lines 115-117).

Line 94: Turbomap – I didn't think this was a Rockland Product. Line 109-116 Turbulence analysis – there are some quite sophisticated and widely used methods for this. Were they used? Various references by Lueck, Wolk and colleagues look at vibration limit identification and replacement of the missing spectral region. Why was 1 m chosen as bin size?

Responses: Thanks for your comments and questions. We removed the 'Rockland Scientific Inc.' in the revised text (line 123). We used an integrated software application TMTools$^{TM}$ developed by Alec Electronics Co., Ltd. to derive the dissipation rate (lines 133-135). 1 m was chosen because the instrument itself swings at low frequency during the free falling process. These low-frequency oscillations would contaminate the low wavenumber region of the spectrum.

Line 122 – "time interval of turbomap"? You mean the individual profiles of the full sampling period?

Responses: Yes, averaged velocity during Turbomap measurement.

**Results**

There are many brief statements here that would better fit in the Discussion in a more expanded form. E.g. lines 161-162, 178-179

Responses: Thanks for your advice. We have moved these statements to the introduction (lines 58-62).

Line 190-191 – better placed into methods.

Responses: Thanks for your advice. We have moved it to the methods (lines 145-146).

Lines 210-on The Gradient Ri gets introduced here which seems strange. It is not

clear why it is required as there exists direct measures of turbulent mixing? Saying that, I do see the point about high vs low shear and stratification.

95   Responses: Thanks for your comments. We deleted this part in the revised text.

**Discussion**

This section is unstructured and would benefit from some clear themes building from the introduction. I think this needs significant work to give it structure and better bring together the results.

100   Responses: Thanks for your advice. We have reconstructed the discussion carefully (lines 301-351).

Lines 260-262 - Instead it starts with some introductory material.

Responses: Thanks for your advice. We have fixed it in the revised text (lines 291-293).

105   Line 260 "chaotic"? Is it actually chaotic or under-sampled? Under-sampling is inevitable in some situations so it is important to be clear. Do we have any sense of nutrient spatial variability beyond the transect data? e.g remote sensing or is the suggestion that these data are of limited use due to the subsurface biological processes?

110   Responses: We apologize for our inappropriate expression. It is not under-sampled. What we want to express is that the distribution of nutrients in transect B is scattered. We have deleted this word to avoid ambiguity (line 290).

We can obtain the sea surface chlorophyll a from ERDDAP (https://coastwatch.pfeg.noaa.gov/erddap/index.html), which is showed in Figure 1. Sea surface chlorophyll a

115   in the region where transect B located was higher than that in the region where transect A located, which suggests that there were more nutrients in the surface layer to support the primary productivity of the region where transect B located. Strong nutrient flux induced by turbulent mixing plays an important role to transport nutrients from deep layer to surface layer.

120   Lines 359-375 - I don't really see the need for separate conclusions especially as they have much in common with abstract.

Responses: Thanks for your comments. We incorporated the conclusions into the discussions (lines 378-395).

Can you plot biological production as a function of the nutrient flux in a way that shows how the two transects differ/compare?

Responses: We cannot estimate biological production due to the lack of data. The relation between Chl-a concentration and $NO_2 + NO_3$ flux is shown in Figure R2. There is no good correlation between Chl-a concentration and $NO_2 + NO_3$ flux. This is expected since nutrient flux is related to the turbulent mixing and the strong nutrient flux mainly occurs in the upper layer, while Chl-a mainly appears in SCML.

[Figure]

**Figure R2**: Chl-a concentration plots against $NO_2 + NO_3$ flux for (a) transect A and (b) transect B.

Can you develop some kind of regime diagram where we have internal wave activity, nutrient availability and wind/upwelling and somehow present your findings for production in these terms? Seasonality gets a minor mention but it would be useful to discuss more fully how these present results could/would translate through the annual cycle.

Responses: We have developed a regime diagram (Figure R3). It is not good as we expect. So we decide not to put it in the text. We have added some discussion about the seasonality in the revised text (lines 385-395).

[Figure]

**Figure R3**: Sketch (not to scale) showing some dynamic processes that are related to nutrient supply.

**Minor**

Potential temperature – some confusing notation and labelling – be good to be clear and use the accepted symbolic notation for potential temperature $\theta_0$.

Responses: Thanks for your advice. We have replaced "$T$" with "$\theta_0$" for potential temperature (line 180).

---

## Author Comment (AC4) · 31 Jul 2020

**Comments**

Thank you for your submission to the Ocean Science Discussion. While the observations you report can be of interest, I find the analysis short of qualifying a scientific article. The manuscript is highly descriptive, and no detailed analysis or convincing discussion is presented. The material would be more appropriate for a (very good) report or a data description paper. Following my comment and the comments from two reviewers, please post an authors' response that outlines how you intend to improve the study (e.g., which additional analysis and revisions you intend to do). At this stage I do not encourage you to prepare a full revised version for submission to Ocean Science, but to summarize how you intend to proceed. Please allow me to decide based on your response.

The observations from two sections (one section with 6 stations and a second section with 9 stations) are presented as distance-depth distributions, like a data report: one figure for hydrography, one figure for the turbulence measurements and one figure for the water samples (Chl-a, nutrients and phosphate concentration). The next figure combines the diffusivity with concentration gradient to obtain the turbulent fluxes. The final figure implies curl-driven upwelling can be important, but it is introduced in passing and not a convincing analysis is presented. Overall, one section is turbulent and the other is not turbulent. Narrative repeatedly compares the two sections in various parameters.

Responses: Thanks for your comment and advice. We have carefully revised the manuscript based on the reviewers' comments. Most of the previous studies on the South China Sea focused on the effects of upwelling, coastal currents, eddies and typhoons on the nutrients and chlorophyll. In this manuscript, we investigated the effects of turbulent mixing on nutrients and chlorophyll. We found the unevenly distributed turbulent mixing has different impacts on the distribution of nutrients and chlorophyll. It is important to our understanding of the impact of turbulent mixing on biological processes, which will draw attention to the biophysics of the South China Sea. Our research is a preliminary study. In the near future, more detailed and longer-term data are needed to study the effects of turbulent mixing on the nutrients,

chlorophyll and primary productivity in space and time.

Some minor comments: The observation period is given as approximately 1 month from late April to late May; however, which days the sections were collected are not stated. If the sections are separated by several weeks, the role of temporal variability could be discussed.

Responses: Thanks for your comment and advice. We have added the date to the text and discussed the role of temporal variability (lines 98-106 and 385-395).

Descriptive occurrence of internal wave packets from satellite images (Zhao et al 2004) are from a different year. More convincing case could be made presenting actual observations from the cruise period.

Responses: Thanks for your comment and advice. The internal wave packets occur every year and are extensive in the northern SCS. To make the article more rigorous, we delete the internal wave packets from Figure 1 and replaced them with sea surface chlorophyll a (line 89).

Please make sure to use same units, for example nitrate fluxes are given in mmol N m-2 s-1 in line 43 and micromole m-2 s-1 in line 46. Throughout, is it correct to refer to CTD as hydrological data?

Responses: Thanks for your advice. We have used the same units in the revised text (lines 47-50). Yes, CTD data are hydrological data.

What acoustic frequency is the ADCP?

Responses: The frequency of ADCP is 38 kHz. We have clarified this in the revised text (lines 115-117).

Li 110, fine scale shear variance is introduced right after the microstructure shear variance. This can be confusing for the reader.

Responses: Microscale shear was collected by TurboMAP and was used to calculate the dissipation rate. Finescale shear variance was calculated with the finescale velocity obtained from shipboard ADCP. We have clarified this in the revised text.

Please use standard notation for potential temperature and potential density anomaly. Please do not use psu for salinity unit. Indicated that it is practical salinity, given on practical salinity scale (no units).

Responses: Thanks for your advice. We have removed the units in the text (line 180).

Li 129-130: salinity layers are identified in brackets in density ranges. This is confusing. At least clarify that the values are density.

Responses: Thanks for your advice. We have clarified the values in the revised text (lines 168-170).

Ri at 16-m vertical scale does not resolve the turbulence processes. It can be removed all together. Citing that previous literature reported Ri from 2m to 16 m vertical scale does not help (li 115).

Responses: Thanks for your advice. We have deleted the content of Ri in the revised text (line 247).

In Fig 8, are the two separate parts around section A and B. I assume these are 3-day averages for days corresponding to the section time periods (i.e. you created a mosaic from two images). If so, please clarify in the caption and in the text.

Responses: Thanks for your advice. We have clarified in the caption and in the text (lines 355 and 362).